# Optimal level activity of matrix metalloproteinases is critical for adult visual plasticity in the healthy and stroke-affected brain

Justyna Pielecka-Fortuna[1]*, Evgenia Kalogeraki[1,2], Michal G Fortuna[3,4], Siegrid Löwel[1]

[1]Department of Systems Neuroscience, Bernstein Focus Neurotechnology, Johann-Friedrich-Blumenbach Institute for Zoology and Anthropology, University of Göttingen, Göttingen, Germany; [2]Göttingen Graduate School for Neurosciences, Biophysics, and Molecular Biosciences, University of Göttingen, Göttingen, Germany; [3]Institute for Neurophysiology and Cellular Biophysics, University Medical Center, Göttingen, Germany; [4]German Primate Center, Göttingen, Germany

**Abstract** The ability of the adult brain to undergo plastic changes is of particular interest in medicine, especially regarding recovery from injuries or improving learning and cognition. Matrix metalloproteinases (MMPs) have been associated with juvenile experience-dependent primary visual cortex (V1) plasticity, yet little is known about their role in this process in the adult V1. Activation of MMPs is a crucial step facilitating structural changes in a healthy brain; however, upon brain injury, upregulated MMPs promote the spread of a lesion and impair recovery. To clarify these seemingly opposing outcomes of MMP-activation, we examined the effects of MMP-inhibition on experience-induced plasticity in healthy and stoke-affected adult mice. In healthy animals, 7-day application of MMP-inhibitor prevented visual plasticity. Additionally, treatment with MMP-inhibitor once but not twice following stroke rescued plasticity, normally lost under these conditions. Our data imply that an optimal level of MMP-activity is crucial for adult visual plasticity to occur.

*For correspondence: jpielec@gwdg.de

**Competing interests:** The authors declare that no competing interests exist.

## Introduction

Neuroplasticity is the ability of the brain to adapt both structurally and functionally to changing patterns of activity induced by the environment or intrinsic factors. In the clinical setting, plasticity is important for tissue repair and neural network rewiring, necessary for recovery and optimal post-injury brain function. The primary visual cortex (V1) is a widely used model region for studying sensory plasticity in young vs. adult brains (*Hofer et al., 2006*; *Espinosa and Stryker, 2012*; *Levelt and Hübener, 2012*). In mammals, V1 consists of a monocular and a binocular zone; neurons in the binocular zone respond to stimulation of both eyes but are dominated by the contralateral eye in rodents (*Dräger, 1975*; *1978*). This ocular dominance (OD) can be modified in an experience-dependent manner, by depriving one eye of pattern vision for several days (known as monocular deprivation or MD), as originally observed by Wiesel and Hubel more than 50 years ago (*Wiesel and Hubel, 1965*). OD-plasticity has become one of the major paradigms for studying cortical plasticity. In standard-cage raised mice, OD-plasticity in binocular V1 is most pronounced in 4-week-old animals; reduced, yet present, in early adulthood; and absent in animals older than 110 days of age (*Lehmann and Löwel, 2008*). In 4-week-old mice, 3-4 days of MD are sufficient to induce a

**eLife digest** When part of the brain becomes damaged as a result of injury or disease – for example, a stroke – other brain regions can sometimes take over from the damaged part. This is one example of a phenomenon called brain plasticity. The strengthening and weakening of connections between neurons that underlies learning and memory is another, less extreme, example of plasticity. While the brain is most plastic during childhood, it remains malleable to some degree throughout life.

The brain's visual system in particular shows robust and predictable plasticity, and so is often used by neuroscientists to study mechanisms behind plasticity. In young rodents, taping one eye shut for a few days causes inputs from that eye to visual areas of the brain to become weaker. Inputs from the open eye meanwhile become stronger, leading to improved vision in the open eye. Such plasticity also occurs in adult rodents, but the eye must be closed for longer to produce an effect.

In young animals, this plasticity depends, in part, on enzymes called matrix metalloproteinases (MMPs). These help to regulate a network of proteins called the extracellular matrix, which provides structural support for cells. Pielecka-Fortuna et al. now provide the first evidence that MMP enzymes also contribute to visual plasticity in adult animals. Blocking the activity of MMPs prevented reorganisation of visual areas of the brains of adult mice in response to eye closure, and prevented vision improvements in the open eye.

However, blocking MMP in adult mice whose brains had been damaged by a stroke had the opposite effect. Whereas stroke normally prevents visual system plasticity in response to eye closure, treatment with a single dose of MMP blocker rescued this plasticity. Strikingly, these benefits were lost if the mice were given two doses of MMP blocker, rather than one.

These experiments show that MMP levels must be within a narrow range to support plasticity. In the healthy adult brain, blocking MMPs impairs plasticity. After stroke, MMP levels are increased and reducing them rescues plasticity. The next challenge is to identify the specific MMP enzymes responsible, and to determine whether these changes can be exploited to improve recovery from stroke.

significant OD-shift towards the open eye (juvenile OD-plasticity), while 7 days of MD are needed in 3-month-old animals (adult OD-plasticity) (*Gordon and Stryker, 1996*; *Sawtell et al., 2003*; *Frenkel and Bear, 2004*; *Mrsic-Flogel et al., 2007*; *Sato and Stryker, 2008*). Although the experimental paradigm is rather similar, molecular mechanisms underlying 'juvenile' and 'adult' OD-plasticity are different: in juvenile mice, OD-shifts are mostly mediated by reductions in deprived eye responses while adult plasticity is predominantly mediated by an increase in open eye responses in V1 (*Hofer et al., 2006*; *Espinosa and Stryker, 2012*; *Levelt and Hübener, 2012*).

Activity driven modifications in neuronal circuits can be facilitated by degradation of the extracellular matrix (ECM) (*Pizzorusso et al., 2002*; *de Vivo et al., 2013*), which provides structural and biochemical support for the cells (*Frischknecht and Gundelfinger, 2012*). Structural and molecular composition of the ECM changes during development, becoming denser and more rigid in the mature brain (*Frischknecht and Gundelfinger, 2012*; *de Vivo et al., 2013*). This structural composition is partially regulated by a family of zinc dependent endopeptidases, the matrix metalloproteinases (MMPs), and their enzymatic activity is crucial for proper development, function and maintenance of neuronal networks (*Milward et al., 2007*; *Huntley, 2012*). A recent study in juvenile rats showed that pharmacological inhibition of MMPs with a broad spectrum inhibitor during the MD-period did not influence the reduction of the closed-eye responses induced by 3 days of MD, yet it prevented the potentiation of the nondeprived eye responses in V1 after 7 days of MD (*Spolidoro et al., 2012*). Whether MMPs are involved in adult OD-plasticity, for which mechanisms are believed to be different (*Hofer et al., 2006*; *Sato and Stryker, 2008*; *Ranson et al., 2012*), is not yet known, and clarifying this point was the first goal of this study.

In addition to MMP involvement in healthy brain function, their excessive activity can also be detrimental (*Agrawal et al., 2008*; *Huntley, 2012*). Both human and animal studies found upregulated activity of MMPs following inflammation, infectious diseases or brain trauma (*Agrawal et al., 2008*; *Rosell and Lo, 2008*; *Morancho et al., 2010*; *Vandenbroucke and Libert, 2014*), and

pharmacological inhibition of MMPs shortly after brain injuries reduced infarct sizes and prompted better recovery (*Romanic et al., 1998*; *Lo et al., 2002*; *Gu et al., 2005*; *Wang and Tsirka, 2005*; *Yong, 2005*; *Morancho et al., 2010*; *Chang et al., 2014*; *Vandenbroucke and Libert, 2014*). Stroke can influence synaptic activities within the area directly affected by it, and also in a broader area surrounding the lesion (*Witte et al., 2000*; *Murphy and Corbett, 2009*). Likewise, impairments in experience-dependent plasticity after a cortical stroke also have been observed in distant brain regions (*Jablonka et al., 2007*; *Greifzu et al., 2011*): in 3-month-old mice, a focal stroke in the primary somatosensory cortex (S1) prevented both V1-plasticity and improvements of the spatial frequency and contrast thresholds of the optomotor reflex of the open eye (*Greifzu et al., 2011*). Interestingly, some MMPs were shown to be upregulated within the first 24 hours after focal stroke (*Cybulska-Klosowicz et al., 2011*; *Liguz-Lecznar et al., 2012*). Thus, the second goal of our study was to test whether balancing the upregulated MMPs resulting from a focal stroke in S1 would rescue visual plasticity.

Using a combination of *in vivo* optical imaging of intrinsic signals and behavioral vision tests in adult mice, we observed that application of the broad spectrum MMP-inhibitor GM6001 during the 7-day MD-period prevented both OD-plasticity and enhancements of the optomotor response of the open eye. In addition, a single treatment after the S1-stroke rescued plasticity in both paradigms, whereas treatment with the inhibitor two times diminished plasticity-promoting effect. Together, these data reveal a crucial role of MMPs in adult visual plasticity and suggest that MMP-activity has to be within a narrow window for experience-induced plasticity to occur.

## Results

### Inhibition of MMPs prevented experience-induced adult visual cortex plasticity

MMPs were shown to be critical for open eye potentiation after 7 days of MD in *juvenile* rats (*Spolidoro et al., 2012*). Since it is believed that the mechanisms underlying juvenile and adult OD-plasticity are different (*Hofer et al., 2006*; *Sato and Stryker, 2008*; *Ranson et al., 2012*), we aimed to test whether MMPs also play a significant role in *adult* V1-plasticity, in which open-eye potentiation is a major component mediating these changes. First, we induced 7-day-MD in 3-month-old mice and examined whether treatment with GM6001 (50mg/kg/day, for 7days), a broad-spectrum MMP-inhibitor, can prevent ocular dominance (OD) plasticity, measured by *in vivo* intrinsic signal optical imaging in V1. *Figure 1* shows representative examples of optically recorded activity and polar maps after visual stimulation of the right (deprived, contralateral) and left (open, ipsilateral) eye in the binocular region of the left V1 in vehicle-treated (control, *Figure 1A,C*) and GM6001-treated adult mice (GM6001, *Figure 1B,D*). In both, control and GM6001-treated mice without MD (*Figure 1A,B*) the activity patch induced by stimulation of the contralateral eye was darker than that of the ipsilateral eye; the OD-index (ODI) was positive and warm colors dominated the OD-map, indicating typical contralateral eye dominance in V1. 7 days of MD induced an OD-shift towards the open eye in control (*Figure 1C*) but *not* in GM6001-treated mice (*Figure 1D*): in control mice after MD visual stimulation of the contralateral and ipsilateral eye activated V1 rather equally strong; the OD-histogram was shifted to the left and cooler colors appeared in the 2-dimensional OD-map. In contrast, V1-activation of GM6001-treated mice after MD was similar to values without MD and the deprived eye still dominated V1-activity (compare *Figure 1B and D*).

Quantification of all imaging data is illustrated in *Figure 2*. In control mice OD-indices were significantly reduced after MD (control, n=6, 0.27 ± 0.03 vs. control+MD, n=5, 0.02 ± 0.01, t-test, p<0.001, *Figure 2A*), and the OD-shift was mediated by an increase in open eye responses in V1 (control, 1.3 ± 0.2 vs. control+MD, 1.8 ± 0.1, t-test, p<0.05, *Figure 2B*), with no concomitant changes in visually driven activity after deprived eye stimulation (control, 1.9 ± 0.2 vs. control+MD, n=5, 1.8 ± 0.1, t-test, p>0.05), indicating a typical *adult* type of OD-plasticity. Thus, the vehicle treatment did not interfere with the induction of experience-dependent cortical plasticity in adult mouse V1. In contrast, in GM6001-treated mice OD-indices were indistinguishable before and after MD (GM6001, n=5, 0.23 ± 0.02 vs. GM6001+MD, n=5, 0.24 ± 0.02, t-test, p>0.05 *Figure 2A*). Likewise, V1-activations via the ipsilateral (GM6001, 1.2 ± 0.2 vs. GM6001+MD, 1.2 ± 0.04) or contralateral eye (GM6001, 1.9 ± 0.2 vs. GM6001+MD, 1.9 ± 0.1) were similar before and after MD (for all, t-test,

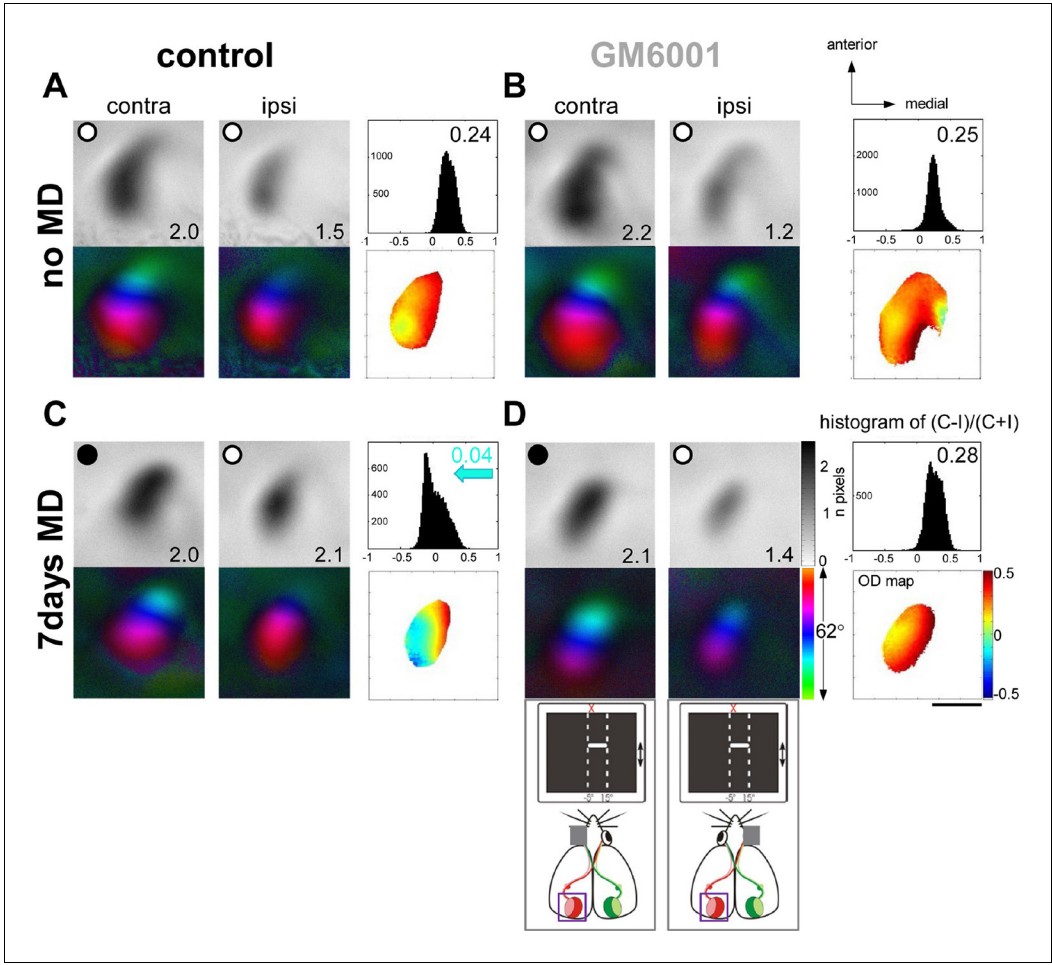

**Figure 1.** Treatment with the MMP-inhibitor GM6001 during MD period prevented OD-plasticity in adult mouse V1. Representative examples of optically recorded activity and polar maps in V1 of both vehicle-treated mice (control, **A, C**) and animals treated with the MMP-inhibitor GM6001 (GM6001, **B, D**), before (no MD, **A, B**) and after monocular deprivation (7 days MD, **C, D**). V1-maps after visual stimulation of the contra- (contra) and ipsilateral (ipsi) eye in the binocular region of V1. Top panels (**A–D**) display grayscale-coded response magnitude maps (V1-activation) and their quantification on the right: histogram of (C-I)/(C+I). V1-activation is illustrated as fractional change in reflection $\times 10^{-4}$. Average V1-activation is illustrated as a number at the lower right corner of each magnitude map; the average OD-index (ODI) as a number in the upper right corner of the histograms. Bottom panels represent color-coded polar maps of retinotopy, and the color-coded OD-map. In both control and GM6001-treated mice without MD, V1-activity was dominated by input from the contralateral eye: activity patches after contralateral eye stimulation were darker than after ipsilateral eye stimulation, the average ODI was positive, and warm colors prevailed in the OD-map, illustrating contralateral dominance. After 7 days of MD, OD-plasticity occurred only in control (**C**) but not in GM6001-treated mice (**D**): in control mice, both eyes activated V1 rather equally strong, the OD-histogram was shifted to the left (blue arrow in **C**), and colder colors appeared in the OD-map. In contrast, in GM6001-treated mice, the deprived (contralateral) eye continued to dominate V1, the ODI was not reduced and warm colors still dominated the OD-map. Scale bar: 1mm.

p>0.05, *Figure 2B*), indicating that the deprived, contralateral eye still dominated V1. This absence of any detectable OD-shift during MMP-inhibition strongly suggests a crucial role of MMPs for mediating visual cortical plasticity in the adult brain.

To test whether injections of GM6001 or vehicle influenced basic properties of V1, we analyzed V1-activation and the quality of the retinotopic maps in all experimental groups without MD. Essentially, there were no significant differences between control (vehicle-treated) and GM6001-treated mice in both map quality and V1-activation (control/GM6001, elevation, map scatter: 0.9 ± 0.2/1.1 ± 0.1; V1-activation: 3.8 ± 0.5/3.2 ± 0.4; azimuth, map scatter: 12.1 ± 2.9/4.8 ± 1.9, V1-activation: 3.1 ± 0.5/2.8 ± 0.3; t-test, p>0.05 for all comparisons), indicating that GM6001/vehicle-treatment did not impact basic map quality nor magnitude of V1-activation.

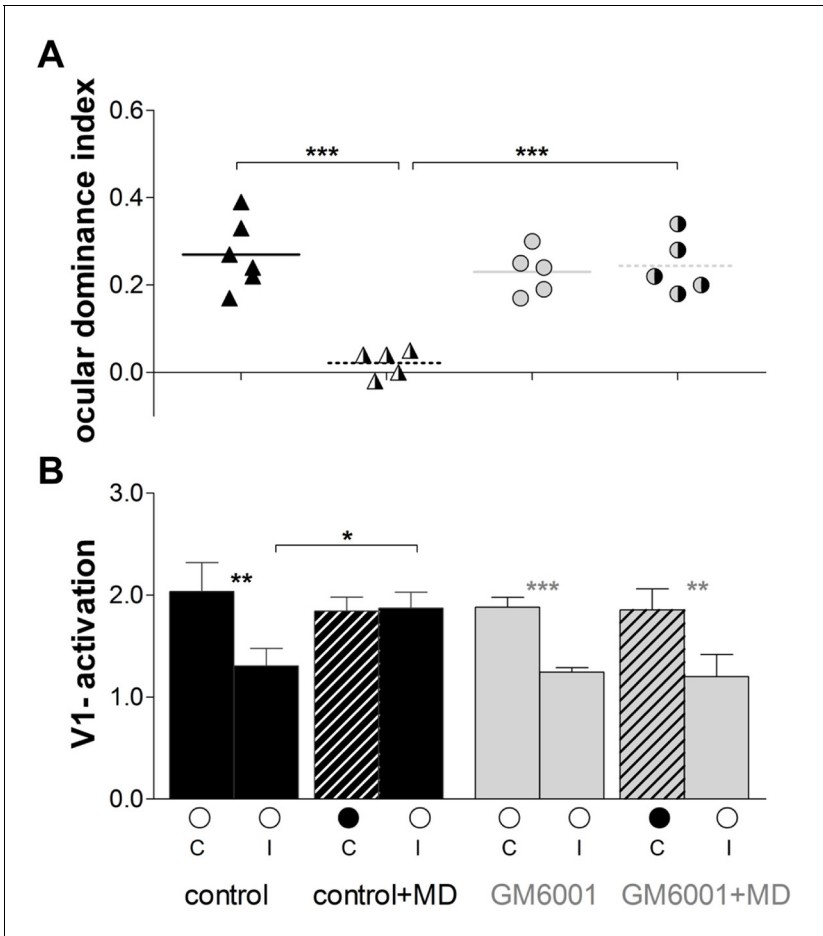

**Figure 2.** Inhibition of MMPs during the MD-period prevented adult OD-plasticity. (**A**) Optically imaged OD-indices in animals without MD (solid symbols) and after 7 days of MD (right half of symbols is black) in control (vehicle-treated, black) and in GM6001-treated mice (grey). Symbols illustrate ODI-values of individual cases; means are marked by horizontal lines. (**B**) V1-activation elicited by stimulation of the contralateral (**C**) or ipsilateral (**I**) eye in animals before (open circles) and after MD (closed eye marked as a black filled circle, and striped bar). The OD-shift in control mice was mediated by an increase of open eye responses in V1, while no OD-shift and no changes in V1-actiavtion were observed in GM6001-treated mice after MD. Mean ± SEM, *p<0.05, **p<0.01, ***p<0.001. ANOVA followed by multiple comparisons with Bonferroni correction was used in **A**; two-tailed t-test was used in **B**.

The following source data is available for figure 2:

**Source data 1.** Ocular dominance index and V1-activation individual values for *Figure 2*.

## Inhibition of MMPs prevented experience-enabled improvements in visual capabilities

In addition to our optical imaging experiments, we also performed behavioral assessments of visual abilities in the same groups of mice. While the experience-induced improvements in the spatial frequency and contrast thresholds of the optomotor reflex of the open eye after MD are cortex-dependent (*Prusky et al., 2006*), the mechanism underlying these enhancements is not yet fully understood. Using the virtual-reality optomotor setup (*Prusky et al., 2004*) and daily testing, we examined the role of MMPs in this kind of interocular plasticity. Spatial frequency and contrast sensitivity thresholds of the optomotor reflex were measured in both vehicle-treated (control) and GM6001-treated mice before and during the 7-day MD-period. Mice without MD did not show any experience-induced changes in the spatial frequency (n=6, day 0: 0.37 ± 0.002 cyc/deg vs. day 7: 0.37 ± 0.001 cyc/deg, t-test, p>0.05, *Figure 3A*) or contrast sensitivity thresholds of the optomotor reflex (day 0 vs. day 7, for all spatial frequencies, ANOVA, p>0.05, *Figure 3B* and *Table 1*). In

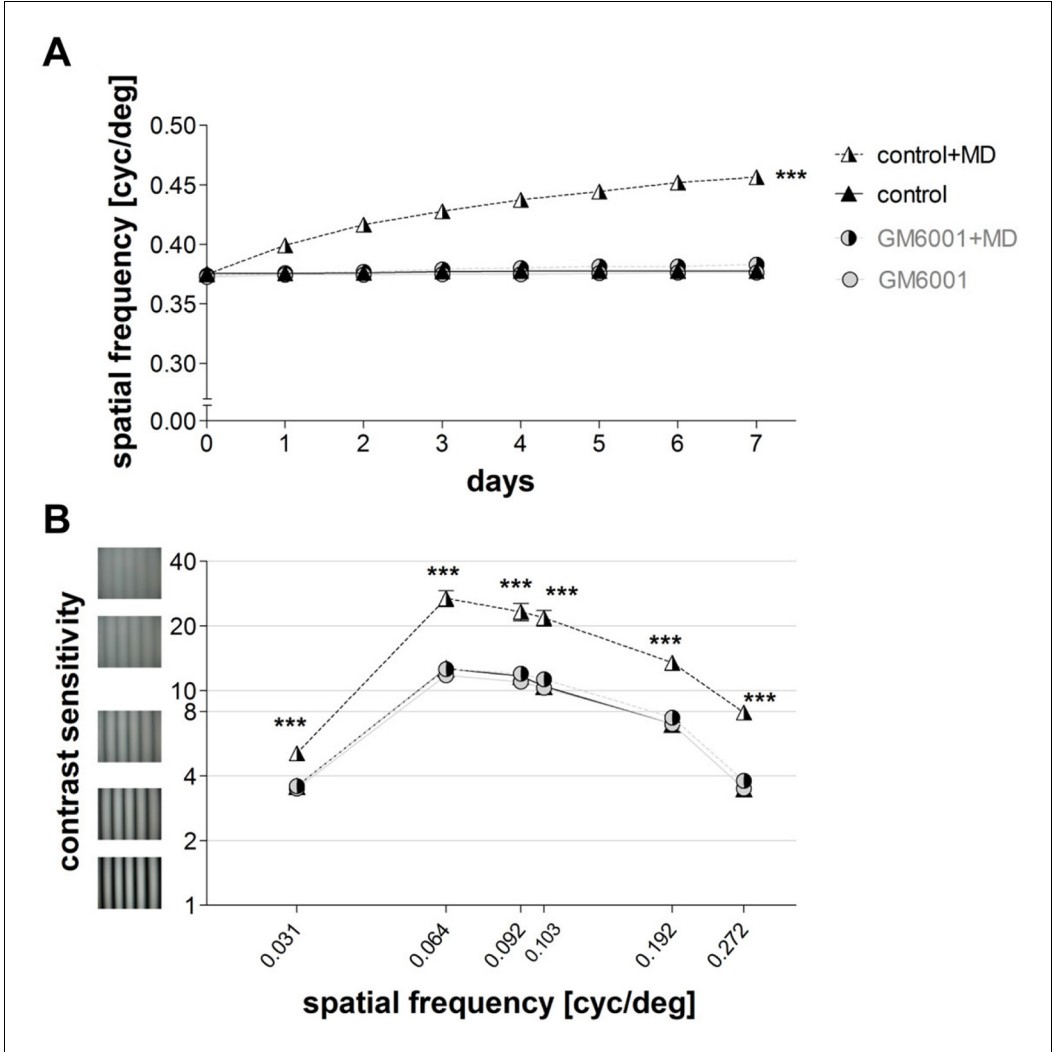

**Figure 3.** Inhibition of MMPs prevented experience-enabled enhancements of both the spatial frequency (**A**) and contrast sensitivity (**B**) thresholds of the optomotor reflex of the open eye in adult mice after MD. Values of vehicle-treated (control) mice are displayed in black (triangles), values of GM6001-treated mice in grey (circles); values of animals with monocular deprivation (+MD) are marked by displaying the right half of the symbol in black. (**A**) Spatial frequency threshold values in cycles/degree (cyc/deg), measured in the optomotor setup plotted against days. (**B**) Contrast sensitivity values on day 7 at 6 different spatial frequencies. Mean ± SEM; ***p<0.001, ns p>0.05. ANOVA followed by multiple comparisons with Bonferroni correction was used; two-tailed t-test was used within the group analysis in **A**.

The following source data is available for figure 3:

**Source data 1.** Optomotry measured spatial frequency and contrast sensitivity thresholds individual values for *Figure 3*.

contrast, vehicle-treated (control) mice with 7 days of MD showed the typical enhancements of both the spatial frequency and contrast thresholds of the optomotor reflex: the spatial frequency threshold of the open eye increased on average by 22 ± 1% from day 0 to day 7 (day 0/7: 0.37 ± 0.002/ 0.46 ± 0.0001 cyc/deg, n=5, t-test, p<0.001, *Figure 3A*) and was thus significantly different from control mice without MD (n=6) that displayed no experience-driven changes (ANOVA, p<0.001, *Figure 3A*). Likewise, contrast sensitivity of the open eye of control mice increased after MD (for all spatial frequencies, ANOVA, p<0.001, *Table 1*), and was different on day 7 compared to control mice without MD (control, n=5 vs. control+MD, n=5, t-test, p<0.001, *Figure 3B*). In contrast, in GM6001-treated mice MD did neither induce an improvement of the spatial frequency (day 0/day 7: 0.37 ± 0.002/0.38 ± 0.002 cyc/deg, n=5, t-test, p>0.05 *Figure 3A*), nor of contrast sensitivity thresholds of the optomotor reflex of the open eye (at all spatial frequencies, ANOVA, p>0.05,

**Table 1.** Optomotry-measured contrast sensitivity improvements after MD

| Spatial frequency (cyc/deg) | Contrast sensitivity | | | |
| --- | --- | --- | --- | --- |
| | Day 0 | | | |
| | Control | GM6001 | Control+MD | GM6001+MD |
| 0.031 | 3.6 ± 0.04 | 3.5 ± 0.02 | 3.5 ± 0.01 | 3.5 ± 0.01 |
| 0.064 | 12.7 ± 0.52 | 11.8 ± 0.16 | 11.6 ± 0.23 | 12.0 ± 0.15 |
| 0.092 | 11.7 ± 0.51 | 10.9 ± 0.18 | 10.7 ± 0.22 | 11.3 ± 0.17 |
| 0.103 | 10.3 ± 0.54 | 10.2 ± 0.10 | 10.0 ± 0.16 | 10.6 ± 0.12 |
| 0.192 | 6.9 ± 0.07 | 6.9 ± 0.12 | 7.0 ± 0.11 | 6.8 ± 0.05 |
| 0.272 | 3.5 ± 0.04 | 3.5 ± 0.02 | 3.5 ± 0.01 | 3.5 ± 0.01 |
| | Day 7 | | | |
| 0.031 | 3.6 ± 0.04 | 3.5 ± 0.02 | 5.1 ± 0.17 | 3.6 ± 0.02 |
| 0.064 | 12.7 ± 0.47 | 11.8 ± 0.18 | 26.9 ± 2.35 | 12.6 ± 0.29 |
| 0.092 | 11.7 ± 0.44 | 11.0 ± 0.15 | 23.3 ± 2.17 | 12.0 ± 0.22 |
| 0.103 | 10.5 ± 0.49 | 10.3 ± 0.10 | 21.8 ± 1.83 | 11.3 ± 0.17 |
| 0.192 | 7.0 ± 0.08 | 7.0 ± 0.10 | 13.5 ± 1.08 | 7.5 ± 0.21 |
| 0.272 | 3.5 ± 0.04 | 3.5 ± 0.01 | 4.9 ± 0.17 | 3.8 ± 0.25 |

*Figure 3B*). Treatment in the no MD groups (control, GM6001) had no effect on baseline spatial frequency and contrast thresholds (for all measurements, ANOVA, p>0.05, *Table 1*, *Figure 3*), demonstrating that MMP-inhibition did not affect the ability to exhibit the optomotor reflex but prevented the experience-enabled changes induced by MD. Together, our behavioral vision tests indicate an essential role for MMPs in this cortex-dependent interocular plasticity paradigm: MMP-inhibition completely abolished the experience-enabled improvements of the optomotor reflex of the open eye.

## Brief inhibition of MMPs rescued experience-induced visual cortex plasticity after stroke

Upregulated activity of MMPs can negatively contribute to the pathology of stroke as well as other neurodegenerative diseases (*Rosenberg et al., 1996*; *Rosenberg, 2002*; *Agrawal et al., 2008*). Furthermore, increased MMP-activity was observed within 24 hours of a photothrombotically (PT) induced stroke, and application of a broad spectrum MMP-inhibitor *at the time of* the stroke partially rescued impaired barrel cortex plasticity (*Cybulska-Klosowicz et al., 2011*; *Liguz-Lecznar et al., 2012*). Here we tested if the stroke-induced impairment of visual cortical plasticity (*Greifzu et al., 2011*) can be rescued by inhibiting MMPs *after* stroke. We examined a total of six groups of mice: all mice received a PT-stroke in their left S1 cortex, about 1 mm anterior to the anterior border of V1. Groups 1 and 2 received vehicle injections 1 h after PT (PT vehicle, PT+MD vehicle), groups 3 and 4 a single injection of the broad spectrum MMP-inhibitor GM6001 1h after stroke (PT 1xGM6001 and PT+MD 1xGM6001 1 h), group 5 received a single injection of GM6001 24 h after stroke (PT+MD+1xGM6001 24 h) and group 6 received 2 injections of GM6001 1 h *and* 24 h after PT (PT+MD 2xGM6001 1 h+24 h). Groups 2, 4, 5 and 6 were additionally subjected to MD.

Histological analysis showed no difference in the volume of the lesion between vehicle- and GM6001-treated mice (vehicle/1xGM6001 1 h/1xGM6001 24 h/2xGM6001 1 h+24 h: 0.6 ± 0.2/1.1 ± 0.3/0.6 ± 0.3/1.2 ± 0.2 mm$^3$, ANOVA, p>0.05). Since there was no difference in the volume of the lesion between mice that received GM6001 once or twice (ANOVA, p>0.05), we pooled these data for further analysis of lesion size and position. The PT-lesion of vehicle-treated mice measured on average 1.1 ± 0.01 mm in medio-lateral and 1.0 ± 0.02 mm in anterior-posterior direction. The lesion center was located 1.2 ± 0.4 mm anterior to the anterior border of V1, 1.7 ± 0.2 mm lateral to the midline, and 1.0 ± 0.3 mm posterior to the Bregma. For GM6001-treated mice, PT-lesion measured on average 1.2 ± 0.09 mm in medio-lateral and 1.0 ± 0.01 mm in anterior-posterior direction. Center of the lesion was situated 0.9 ± 0.3 mm anterior to the anterior border of V1, 1.9 ± 0.1 mm lateral to

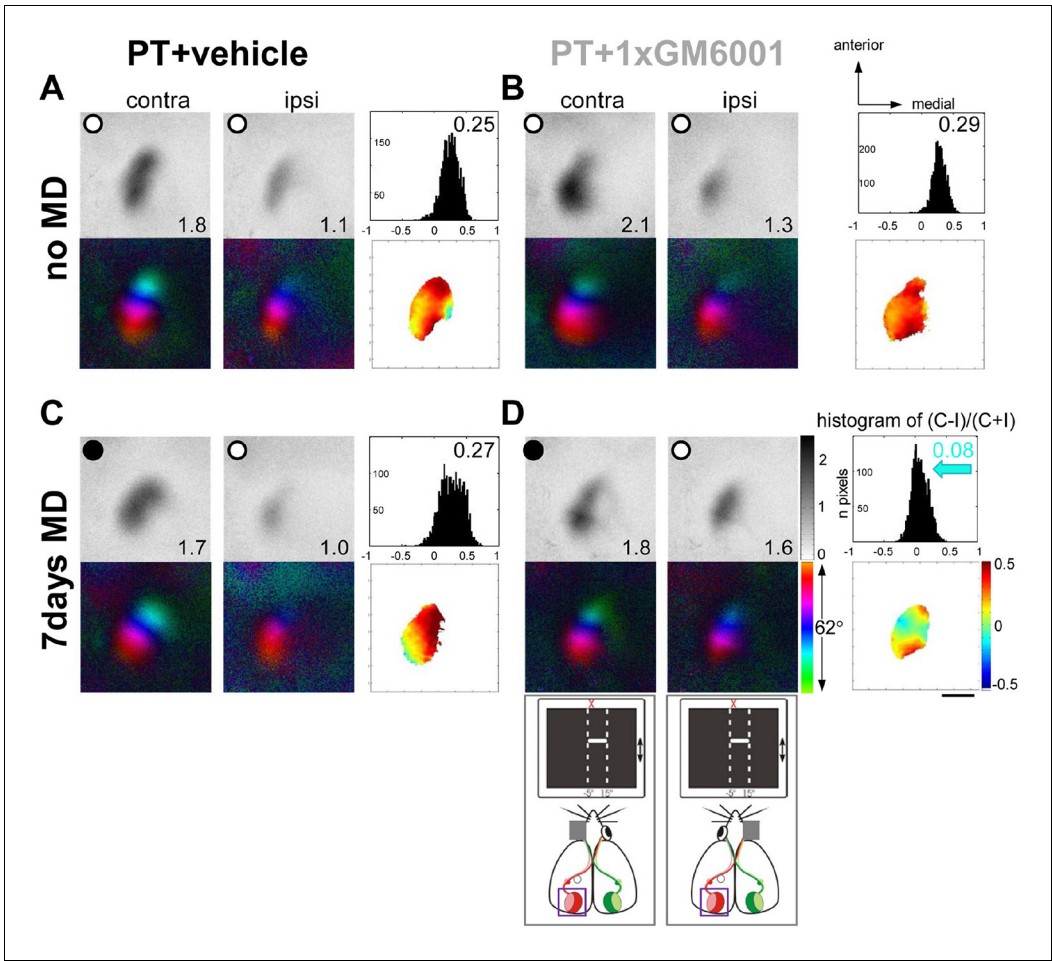

**Figure 4.** A single GM6001-treatment rescued OD-plasticity in V1 after a stroke in S1. Representative examples of optically recorded activity and polar maps in V1 of both vehicle- (PT+vehicle, **A, C**) and single GM6001-treated mice (PT+1xGM6001, **B, D**), before (no MD, **A, B**) and after 7 days of MD (7days MD, **C, D**). V1-maps after visual stimulation of the contra- (contra) and ipsilateral (ipsi) eye in the binocular region in V1. Figure layout and data display as in *Figure 1*. In both vehicle- (**A**) and GM6001-treated PT-lesioned mice without MD (**B**), V1 activation was dominated by the contralateral eye: activity patches induced by contralateral eye stimulation were darker than after ipsilateral eye stimulation, the average ODI was positive, and warm colors dominated the OD-map. (**C**) While 7 days of MD did not induce an OD-shift in vehicle-treated PT mice, and deprived (contra) eye patches still dominated V1, a single treatment with GM6001 rescued OD-plasticity in PT mice (**D**): both eyes activated V1 about equally strong, the histogram of ODIs shifted to the left (blue arrow) and colder colors appeared in the OD-map. Scale bar: 1mm

the midline, and 1.2 ± 0.2 mm posterior to the Bregma. There was no significant difference in the location nor the size of the lesion between vehicle- and GM6001-treated mice (ANOVA, p>0.05 for every measurement).

*Figure 4* shows representative examples of optically recorded activity and polar maps in V1 of adult S1-lesioned mice before and after MD. In both vehicle- and GM6001-treated mice without MD (*Figure 4A,B*), the activity patch induced by stimulation of the contralateral eye appeared darker than that of the ipsilateral eye, the ODI was positive and warm colors dominated in the OD-map, indicating that treatment with the MMP-inhibitor or vehicle after the PT-lesion did not influence baseline ocular dominance and sensory-induced activity in binocular V1 (*Figure 4A,B*). As previously observed in untreated mice, MD after small S1-lesions was also not able to induce an OD-shift towards the open eye in vehicle-treated mice (*Figure 4C*): V1-activity of PT-lesioned mice after MD was almost indistinguishable from animals without MD (compare *Figure 4A and C*), and V1 remained dominated by the deprived, contralateral eye. In contrast, GM6001-treatment rescued the OD-shift, since after MD V1 was activated equally strongly by both eyes, the ODI-histogram shifted to the left and colder colors appeared in the OD-map (*Figure 4D*).

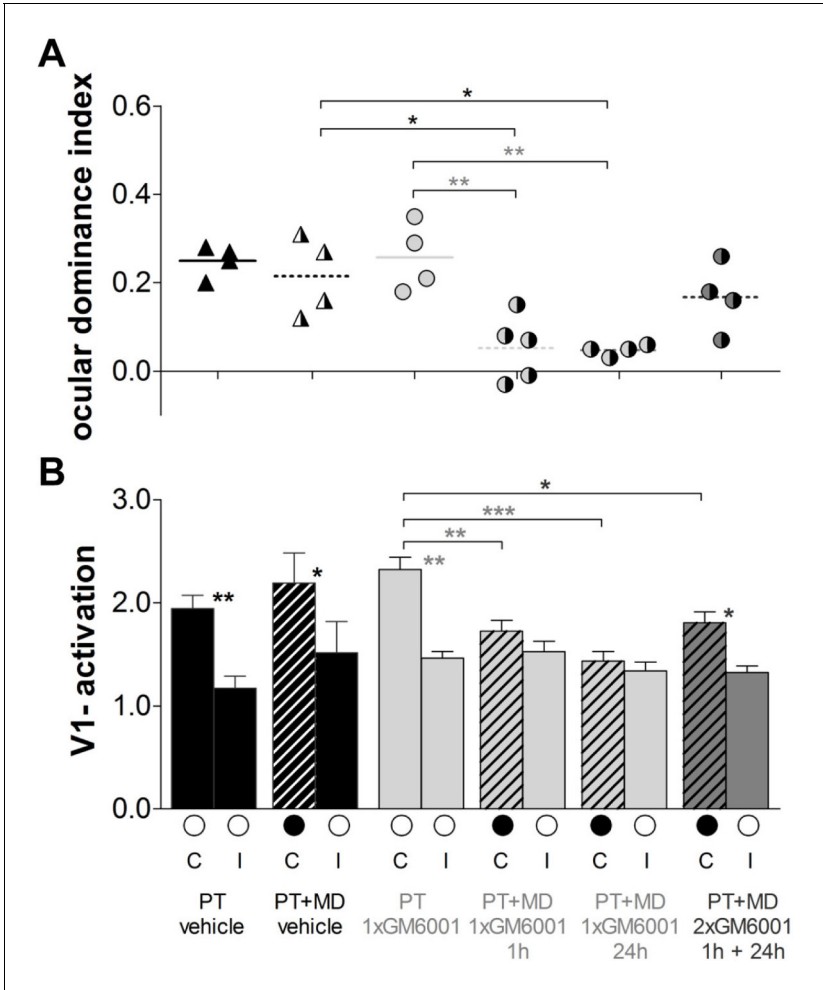

**Figure 5.** Brief MMP-inhibition after a stroke in S1 rescued experience-dependent plasticity in V1. Quantification of the imaging data, layout and data display as in *Figure 2*. (A) Optically imaged OD-indices in mice without MD and after 7 days of MD in vehicle-treated 1h after PT (black) mice and single GM6001-treated (PT 1xGM6001; PT+MD 1xGM6001 1 hr and PT+MD 1xGM6001 24 h, light grey) or two-times GM6001-treated (PT+MD 2xGM6001 1 h+24 h, dark grey) after PT mice. (B) V1-activation elicited by stimulation of the contralateral (C) or ipsilateral (I) eye in animals before (open circles) and after MD. V1-activation did not change after MD in vehicle-treated PT mice. In contrast, 1xGM6001 (either 1 h or 24 h) but not 2xGM6001 (1 h+24 h) treatment rescued OD-plasticity after the PT-lesion (A). There was a significant reduction in deprived eye responses in V1 in the GM6001- but not vehicle-treated mice after MD (B). Mean ± SEM, *p<0.05, **p<0.01, ***p<0.001. ANOVA followed by multiple comparisons with Bonferroni correction was used; two-tailed t-test was used within the group analysis in B.

The following source data is available for figure 5:

**Source data 1.** Ocular dominance indexes and V1-activation individual values for *Figure 5*.

Quantification of all individual cases is presented in *Figure 5*. In vehicle-treated mice, OD-plasticity was abolished after PT (PT vehicle, n=4, 0.25 ± 0.02 vs. PT+MD vehicle, n=4, 0.22 ± 0.04, t-test, p>0.05, *Figure 5A*) and there was no change in visually driven activity in V1 from either the deprived (C: PT vehicle, 1.9 ± 0.1 vs. PT+MD vehicle, 2.2 ± 0.3, t-test p>0.05) or the open eye (I: PT, 1.2 ± 0.1 vs. PT+MD, 1.5 ± 0.3, t-test, p>0.05) (*Figure 5B*). In contrast, treatment once, but not twice, with GM6001 rescued OD-plasticity after stroke. Specifically, mice which received a single injection of GM6001 1 h or 24 h after PT-induction showed a significant OD-shift, while mice receiving two injections (1 h+24 h) after the injury did not display OD-plasticity (PT+GM6001, n=5, ODI=0.26 ± 0.04 vs. PT+MD+1xGM6001 1 h, n=5, 0.05 ± 0.04, p<0.01; vs PT+MD+1xGM6001 24 h, n=4, 0.05 ± 0.01, p<0.01 and vs. PT+MD+2xGM6001 1 h+24 h, n=4, 0.17 ± 0.04, p>0.05, ANOVA *Figure 5A*). Interestingly, in the three MD groups treated with GM6001, deprived eye responses were reduced

compared to GM6001-treated mice without MD (PT+GM6001, 2.3. ± 0.1 vs. PT+MD+1xGM6001 1 h, 1.7 ± 0.1, ANOVA, p<0.01; vs PT+MD+1xGM6001 24 h, 1.4 ± 0.1, p<0.001; and vs. PT+MD +2xGM6001 1 h+24 h, 1.8 ± 0.1, p<0.05, ANOVA) and there was no change in open eye responses in V1 (PT+GM6001, 1.5 ± 0.1 vs. PT+MD+1xGM6001 1 h, 1.5 ± 0.1; vs PT+MD+1xGM6001 24 h, 1.3 ± 0.1; and vs. PT+MD+2xGM6001 1 h+24 h, 1.3 ± 0.1, for all p>0.05, ANOVA), indicating strongly reduced but not entirely absent plasticity-promoting effect in the two-days treatment group (*Figure 5B*).

## Inhibition of MMPs after induction of a cortical lesion rescued experience-induced improvements in visual capabilities in adult mice

We have previously shown that even a small PT-lesion in S1, i.e. close to but not within V1, can prevent the experience-induced enhancement of the optomotor reflex of the open eye after MD (*Greifzu et al., 2011*): the S1-lesion abolished improvements in both the spatial frequency and contrast sensitivity thresholds of the optomotor reflex. Since upregulated MMPs can have a detrimental effect on experience-dependent plasticity (*Cybulska-Klosowicz et al., 2011*), and thus on recovery from brain injuries (*Rosell and Lo, 2008*; *Yang et al., 2013*; *Chaturvedi and Kaczmarek, 2014*), here we tested whether a brief inhibition of MMPs after a cortical stroke can also rescue lost optomotor enhancement of the open eye after MD.

In vehicle-treated mice, MD did not induce any improvements in the spatial frequency thresholds of the optomotor reflex of the open eye (PT+MD vehicle): on day 0, acuity was 0.38 ± 0.002 cyc/deg and 0.38 ± 0.004 cyc/deg on day 7 (n=4, t-test, p>0.05, *Figure 6A*); values were not different from vehicle-treated mice without MD (PT vehicle, n=4, vs. PT+MD vehicle, n=4, p>0.05, *Figure 6A*). Thus, visual improvements after MD in vehicle-treated animals were as compromised as previously described for wild type mice after PT (*Greifzu et al., 2011*). Similarly, we did not observe any improvements in contrast sensitivity thresholds of the optomotor reflex at all spatial frequencies (n=4, ANOVA, for all p>0.05, *Table 2*), and values were similar for mice with or without MD (PT vehicle, n=4 vs. PT+MD vehicle, n=4, p>0.05, *Figure 6B*).

As there was no significant difference in the spatial frequency (ANOVA, p>0.05) or contrast sensitivity thresholds of the optomotor reflex (ANOVA, p>0.05) between mice receiving a single or two injections of GM6001, we pooled data for analysis. In contrast to vehicle-treated, PT-lesioned mice receiving a single or two-times injections of the MMP-inhibitor (PT+MD+GM6001) displayed the typical experience-enabled enhancements in the spatial frequency threshold of the optomotor reflex of the open eye after MD: values increased from 0.38 ± 0.002 on day 0 to 0.45 ± 0.003 on day 7 after MD (n=14, t-test, p<0.001). GM6001-treated PT-lesioned mice also improved contrast sensitivity thresholds of the optomotor reflex of the open eye at all measured spatial frequencies (n=14, ANOVA, for all p<0.001, *Table 2*), and exhibited significant increases in contrast sensitivity on day 7, when compared to mice without MD (PT+GM6001, n=4, vs. PT+MD+GM6001, n=14, ANOVA, p<0.001, *Figure 6B*). Altogether, these data show that brief inhibition of MMPs *after* the induction of a cortical stroke lesion completely rescued experience-induced enhancements in both the spatial frequency and contrast sensitivity thresholds of the optomotor reflex of the open eye after MD. This finding also suggests that upregulated MMPs in the immediate post-injury period can have a detrimental influence on interocular plasticity in adult mice.

## Discussion

The objective of this study was to examine if MMPs are crucial for adult visual plasticity, and if inhibition of their upregulation following cortical stroke may be beneficial for rescuing lost plasticity. A combination of *in vivo* optical imaging and behavioral vision tests revealed that an optimal level of MMP-activity is essential for adult visual plasticity to occur in the healthy *and* stroke-affected brain. In healthy adult mice, MMP-inhibition with GM6001 prevented both ocular dominance plasticity and experience-driven improvements of the optomotor reflex of the open eye after MD, indicating that MMP-activation is required for normal adult plasticity. In addition, blockade of elevated MMP-activity after a cortical stroke rescued compromised plasticity. Specifically, a single but not two-times treatment with GM6001 after a cortical PT-lesion in the neighboring S1 region fully rescued experience-dependent ocular dominance plasticity in adult V1, which is normally lost under these conditions. These observations suggest that MMP-activity has to be within a narrow window to allow

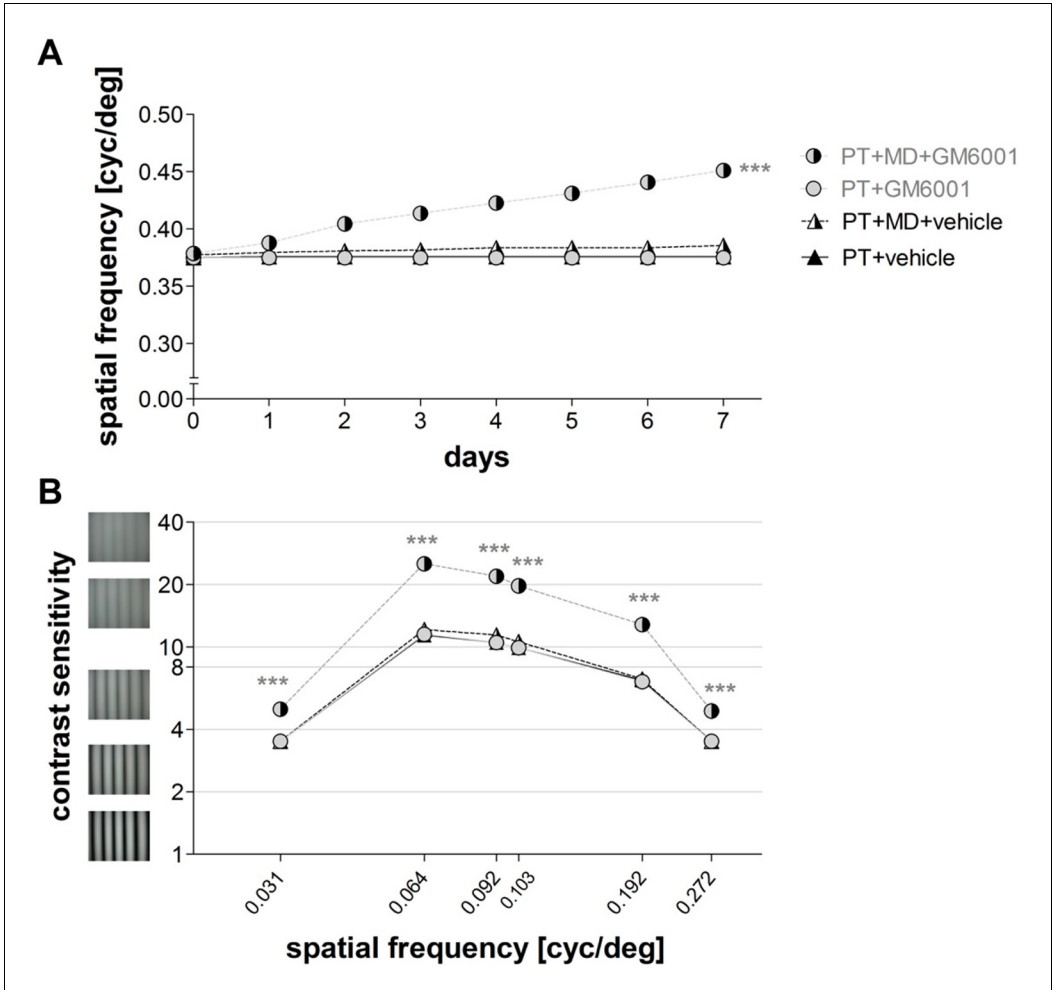

**Figure 6.** Brief inhibition of MMPs after induction of a photothrombotic lesion in S1 rescued experience-enabled enhancements of both the spatial frequency (**A**) and contrast sensitivity (**B**) thresholds of the optomotor reflex in adult mice. Values of vehicle-treated PT-mice are shown in black and values of GM6001-treated mice in grey. Layout and data presentation as in *figure 3*. Mean ± SEM, ***p<0.001, ns p>0.05. ANOVA followed by multiple comparisons with Bonferroni correction and two-tailed t-test was used in A within the group analysis.

The following source data is available for figure 6:

**Source data 1.** Optomotry-measured the spatial frequency and contrast sensitivity individual values for *Figure 6*.

visual plasticity: if MMP-activity is downregulated (with inhibitors) or upregulated (after stroke) experience-induced plastic changes are compromised.

In a healthy brain, the enzymatic activity of MMPs has plasticity promoting effects (*Milward et al., 2007*; *Huntley, 2012*). Consequently to increased neuronal activity, activated MMPs can lessen physical barriers (loosen up the ECM) or via the degradation of certain receptors influence signaling pathways. Such changes within neuronal cells and their synapses thereby alter the structure and activity of neuronal networks (*Milward et al., 2007*; *Huntley, 2012*). Involvement of MMPs in synaptic circuit remodeling has been mainly studied in the hippocampus, yet, their abundant expression in the central nervous system (CNS) indicates a much broader function. For instance, MMP9-deficient mice showed modestly reduced experience-dependent plasticity in the barrel cortex (*Kaliszewska et al., 2012*) and MMP3-deficient mice displayed abnormal neuronal morphology in the visual cortex and impaired plasticity induced by long-term monocular enucleation (*Aerts et al., 2014*). Here, we used a different approach to probe the role of MMPs in experience-induced changes in the visual system: we performed the experiments in wild type mice and treated them with the broad-spectrum inhibitor GM6001 or vehicle during 7 days of MD. Consistent with previous

**Table 2.** Optomotry-measured contrast sensitivity improvements after MD

| Spatial frequency (cyc/deg) | Contrast sensitivity | | | |
| --- | --- | --- | --- | --- |
| | Day 0 | | | |
| | PT+vehicle | PT+GM6001 | PT+MD vehicle | PT+MD GM6001 |
| 0.031 | 3.5 ± 0.02 | 3.5 ± 0.04 | 3.5 ± 0.02 | 3.5 ± 0.02 |
| 0.064 | 11.4 ± 0.35 | 11.4 ± 0.32 | 11.8 ± 0.57 | 11.8 ± 0.15 |
| 0.092 | 10.5 ± 0.36 | 10.5 ± 0.34 | 10.7 ± 0.39 | 11.0 ± 0.16 |
| 0.103 | 9.8 ± 0.26 | 9.9 ± 0.28 | 10.0 ± 0.37 | 10.0 ± 0.32 |
| 0.192 | 6.9 ± 0.12 | 6.7 ± 0.09 | 6.7 ± 0.13 | 6.5 ± 0.26 |
| 0.272 | 3.5 ± 0.03 | 3.5 ± 0.04 | 3.4 ± 0.01 | 3.5 ± 0.02 |
| | Day 7 | | | |
| 0.031 | 3.5 ± 0.02 | 3.5 ± 0.04 | 3.5 ± 0.01 | 5.0 ± 0.11 |
| 0.064 | 11.4 ± 0.39 | 11.5 ± 0.32 | 12.1 ± 0.42 | 25.2 ± 1.53 |
| 0.092 | 10.5 ± 0.36 | 10.5 ± 0.34 | 11.4 ± 0.28 | 21.9. ± 1.14 |
| 0.103 | 9.9 ± 0.24 | 9.9 ± 0.28 | 10.6 ± 0.26 | 19.7 ± 1.25 |
| 0.192 | 6.9 ± 0.11 | 6.8 ± 0.07 | 7.0 ± 0.02 | 12.8 ± 0.91 |
| 0.272 | 3.5 ± 0.02 | 3.5 ± 0.04 | 3.5 ± 0.02 | 4.9 ± 0.12 |

findings (*Gordon and Stryker, 1996*; *Sawtell et al., 2003*; *Sato and Stryker, 2008*), the OD-shift of vehicle-treated mice was mediated by open-eye potentiation. In contrast, there was no change in the open nor in the closed eye responses and hence no OD-plasticity in V1 after MD in GM6001-treated adult mice. This is in line with recent observations from juvenile rats, in which chronic treatment with GM6001 also prevented open eye potentiation after 7 days of MD (*Spolidoro et al., 2012*). However, treatment in this study only partially prevented the OD-shift, as there was no effect on the reduction of deprived eye responses (*Spolidoro et al., 2012*). A reduction in deprived eye responses in V1 is mostly observed in juvenile rodents after 3-4 days of MD (*Gordon and Stryker, 1996*), unless different raising conditions are used such as enriched environment or running wheel (*Greifzu et al., 2014*; *Kalogeraki et al., 2014*). In adult, standard-cage raised mice, 6-7 days of MD are necessary for significant OD-shifts and changes are mainly mediated by increases of open eye responses in V1 (*Gordon and Stryker, 1996*; *Sawtell et al., 2003*; *Sato and Stryker, 2008*). MMP9 activity was suggested in the potentiation of the open eye responses in juvenile rats, as treatment with GM6001 significantly reduced MMP9-mRNA expression only in the hemisphere where structural changes took place (*Spolidoro et al., 2012*). Since we observed a full blockade of plasticity after GM6001-treatment, it would be of interest to determine whether MMP9 is crucial for open-eye potentiation also in the adult brain. MMP9 has been widely investigated in various plasticity paradigms (*Milward et al., 2007*; *Frischknecht and Gundelfinger, 2012*; *Huntley, 2012*; *Tsilibary et al., 2014*) and one of the molecules shown to stimulate MMP9 secretion and expression *in vitro and in vivo* is Tumor Necrosis Factor alpha (TNFalpha) (*Hanemaaijer et al., 1993*; *Candelario-Jalil et al., 2007*). TNFalpha signaling was found to play an important role in the open eye potentiation in juvenile (*Kaneko, et al., 2008*), but not in adult V1 plasticity (*Ranson et al., 2012*), and since our data reveal that MMPs are indispensable for adult V1 plasticity, MMP-activation in the adult brain is most likely not dependent on TNFalpha signaling. Accordingly, this adds to the notion that juvenile and adult V1 plasticity depend on different molecular mechanisms (*Hofer et al., 2006*; *Sato and Stryker, 2008*; *Ranson et al., 2012*). Together, our new data demonstrate a vital role of MMPs for adult visual cortex plasticity, in particular for the increase of open eye responses in V1 after MD, and notably expand the previous studies from juvenile rats.

Under normal conditions, MMP-activity supports healthy brain development and function; yet a different outcome of MMP action has been described for diseased brain (*Agrawal et al., 2008*). Under pathophysiological conditions like inflammation, infection or stroke, uncontrolled MMP driven proteolysis can lead to negative consequences (*Rosenberg et al., 1996*; *Rosenberg, 2002*;

*Agrawal et al., 2008*). Excessive MMP-activity after stroke caused blood brain barrier disruption, upregulation of inflammatory mediators, excitotoxicity and eventually cell death (*Romanic et al., 1998*; *Lo et al., 2002*; *Gu et al., 2005*; *Wang and Tsirka, 2005*; *Yong, 2005*; *Morancho et al., 2010*; *Chang et al., 2014*; *Vandenbroucke and Libert, 2014*). Recent studies reported increased enzymatic MMP9 activity within 24 h after a PT-stroke, and application of a broad spectrum MMP-inhibitor (FN-439) applied *at the time* of stroke induction, partially rescued impaired barrel cortex plasticity (*Cybulska-Klosowicz et al., 2011*; *Liguz-Lecznar et al., 2012*). Here we tested whether inhibition of upregulated MMP-activity (resulting from PT-stroke) may rescue cortical plasticity also when the treatment starts *after* lesion induction. Indeed, a single GM6001-treatment after PT in the neighboring S1 fully rescued plasticity in both of our experimental paradigms, OD- and interocular plasticity. Importantly, successful treatment did not have to start immediately after stroke induction (1 h) because treatment 24 h after had the same beneficial effect, highlighting the therapeutic potential of brief MMP-inhibition for stroke recovery. Beneficial treatment was, however, dependent on the number of injections: a single but not two-times injection of the MMP-inhibitor rescued OD-plasticity. The reduced plasticity-promoting effect in V1 of mice treated twice with GM6001 is likely due to *too* excessive MMP-downregulation, which in turn interfered with MMP facilitation of MD-induced plasticity. Consistent with this interpretation, it was reported that a particular dosage, timing as well as duration of MMPs-inhibition mattered for reduciton of lesion sizes after intracerebral hemorrhage, blood brain barrier permability or neurovascular remodeling in post-stroke period (*Wang and Tsirka, 2005*; *Zhao et al., 2006*; *Sood et al., 2008*; *Chang et al., 2014*). Together with the results from healthy animals, our data suggest that the plasticity-promoting effects of MMPs are dependent on a well-balanced level of activation, and if that balance is disturbed, experience-induced plastic changes are compromised.

There are several plausible mechanisms by which MMP-inhibition rescues OD-plasticity after stroke in S1. Stroke influences brain function in numerous ways e.g., causing inflammation and apoptosis, or disrupting the tightly regulated balance of neuronal inhibition and excitation (*Neumann-Haefelin et al., 1995*; *Schiene et al., 1996*; *Witte and Stoll, 1997*; *Que et al., 1999a*; *1999b*; *Witte et al., 2000*) also in perilesional areas (*Murphy and Corbett, 2009*). One of the major consequences of ischemic damage is an elevated level of the neurotransmitter glutamate, leading to excitotoxicity and neuronal death (*Lai et al., 2014*). On the other hand, stroke can lead to increased tonic inhibition in the peri-infarct zone, and reducing this inhibition can promote functional recovery (*Clarkson et al., 2010*). In addition, focal ischemia can induce spreading depression within ipsilateral cortex (*Schroeter et al., 1995*) and a recent study showed that this phenomenon upregulated MMPs, leading to a breakdown of the blood brain barrier, edema, and vascular leakage, which was suppressed by GM6001 treatment (*Gursoy-Ozdemir et al., 2004*). Thus, it is likely that treatment with GM6001 shortly after the stroke (as in the present study) downregulated MMPs, thus reduced spreading depression, improved disturbed excitation/inhibition balance and allowed plastic changes to occur.

Although, we observed clear functional rescue of OD-plasticity after GM6001-treatment, there was no apparent effect on the lesion size: the PT-lesions in GM6001-treated mice were not smaller compared to vehicle-injected mice. This is in line with a recent observation, that a different broad-spectrum MMP-inhibitor (FN-143) did not attenuate brain damage resulting from photothrombosis, but partially rescued barrel cortex plasticity (*Cybulska-Klosowicz et al., 2011*). The present results, together with the above mentioned study, are not easy to reconcile with findings where MMP-inhibitors reduced the volume of a brain injury (*Gu et al., 2005*; *Wang and Tsirka, 2005*; *Vandenbroucke and Libert, 2014*). The difference might arise from the method used for stroke induction, dosage of inhibitors, timing and duration of the treatment and severity of the lesion.

Behavioral vision tests additionally revealed an involvement of MMPs for interocular plasticity during MD. The optomotor reflex is known to be mediated by subcortical pathways (*Giolli et al., 2006*). While visual capabilities measured by optometry mainly reflect the properties of the retinal ganglion cells that project to these subcortical structures (*Douglas et al., 2005*), daily testing in the optomotor setup after MD induces a cortex-dependent and experience-enabled enhancement of spatial vision through the open eye (*Prusky et al., 2006*). Although inflammation was shown to interfere with the experience-enabled optomotor changes (*Greifzu et al., 2011*), little is known about the cellular origins or signaling pathways responsible for this plasticity paradigm. Here, we found that daily application of the MMP-inhibitor during MD prevented enhancements in both the spatial frequency

and contrast sensitivity thresholds of the optomotor reflex of the open eye, while vehicle-treated control mice displayed the typical experience-enabled optomotor improvements. On the other hand, treatment of mice with the same inhibitor once or twice following cortical stroke rescued the impaired optomotor improvements. Thus, in contrast to the OD-plasticity results, rescue of optomotor improvements was present regardless of the duration of the treatment, adding to the idea that separate mechanisms and different neuronal circuits are responsible for OD- and interocular plasticity (*Greifzu et al., 2011*; *2014*; *Kang et al., 2013*). Together, our results establish a novel function of MMPs in experience-enabled enhancements of the optomotor reflex of the open eye after MD in adult mice.

In conclusion, our present data highlight a critical role of MMPs in adult visual plasticity. They further suggest that upregulation of MMP-activity shortly after a cortical lesion compromises experience-induced visual plasticity, which in turn can be rescued by brief MMP-inhibition. Precise regulation of MMP-activity therefore seems to be essential for facilitating plasticity in the adult brain and offers new opportunities in treatment and recovery after stroke. It remains to be determined which particular MMPs account for the present results.

## Materials and methods

All experimental procedures were approved by the local government (Niedersächsisches Landesamt für Verbraucherschutz und Lebensmittelsicherheit, registration number 33.9-42502-04-10/0326).

### Animals

Male adult (n=49, P77-91) wild-type C57Bl/6J mice housed in standard cages (26×20×14 cm) on a 12 h light/dark cycle with food and water available ad libitum were used.

### Behavioral vision tests

To assess basic visual capabilities and experience-induced changes of all experimental animals, we measured both the spatial frequency threshold and contrast thresholds of the optomotor reflex using a virtual-reality optomotor system (*Prusky et al., 2004*). Briefly, freely moving mice were positioned on a small platform surrounded by four computer monitors forming a square. A rotating virtual cylinder covered with vertical sine wave gratings (with a drift speed of 12°/s) were projected on the monitors. The mice reflexively tracked the gratings by head movements as long as they could see the visual stimuli. Spatial frequency at full contrast and contrast at six different spatial frequencies [0.031, 0.064, 0.092, 0.103, 0.192, 0.272 cycles/degree (cyc/deg)] were varied by the experimenter until the threshold of tracking was determined. Contrast sensitivity thresholds measured in percent were converted into Michelson contrasts as described previously (*Goetze et al., 2010*). To stimulate experience-enabled enhancement of visual capabilities behavioral testing was performed daily for 7 days in all mice, starting before MD (day 0) in the MD-groups.

### Monocular deprivation

To induce plasticity and to study the influence of a stroke on experience-dependent plasticity we used monocular deprivation paradigm (*Gordon and Stryker, 1996*). The right eye was closed as described previously (*Greifzu et al., 2011*). Mice were anesthetized with 2% isoflurane in a mixture of $O_2/N_2O$ (75/25%), their eye-lids were sutured, and mice returned to their home cages for recovery. For stroke experiments MD was performed directly after surgery for induction of a photothrombotic stroke. Animals were checked daily to ensure that the eye remained closed. Mice in which the eye was not closed completely were excluded from the experiment.

### MMP-inhibitor administration

The broad spectrum MMP-inhibitor GM6001 (USBiological, Swampscott, Massachusetts) was used and prepared as previously described with slight modifications (*Gursoy-Ozdemir et al., 2004*; *Wang and Tsirka, 2005*; *Chen et al., 2009*). Specifically, mice were injected intraperitoneally with either 50 mg/kg GM6001 diluted in 3% DMSO and 2% cyclodextrin in saline to a total volume of 200 µl, or with 200 µl vehicle (3% DMSO and 2% cyclodextrin in saline). This type of systemic application of GM6001 was used before and shown to successfully inhibit MMPs activity in the brain

tissue (*Gursoy-Ozdemir et al., 2004*; *Wang and Tsirka, 2005*; *Chen et al., 2009*). For the healthy brain group, the injections were applied daily for 7 days, starting 1 h after MD/no MD. For the stroke group, treatment started 1 h after stroke induction, with an additional injection 24 h after stroke in one group of animals.

## Induction of a photothrombotic stroke

The cortical stroke was induced by photothrombosis (PT) in the left primary somatosensory cortex (S1) using the Rose Bengal technique (*Watson et al., 1985*) as described previously (*Greifzu et al., 2011*). The PT-technique was chosen because it allows small lesions with a reproducible localization, which is particularly relevant when studying post-lesion changes in specific cortical areas in proximity of the lesion. Briefly, mice were box-anesthetized with 2% isoflurane in a mixture of 75/25% of $O_2$/$N_2O$; during surgery, anesthesia was maintained with 0.8-1% isoflurane delivered via an inhalation mask. The body temperature was maintained at 37°C via a heating pad. The animals' head was placed in a stereotaxic frame. The skin above the skull was incised and an optic fiber bundle (aperture: 1.0 mm) mounted on a cold light source (Schott KL 1500, Germany) was positioned 2 mm lateral to the midline and 1 mm posterior to the bregma. 100 µl Rose Bengal (0.1% in normal saline) dye was injected into the tail vein. 5 min after the dye injection, the illumination (lasting 15 min) began. The skin above the skull was sutured and animals returned to their home cages for recovery. Lesion size was determined at the end of the behavioral and optical imaging experiments by Nissl staining.

## Optical imaging of intrinsic signals and visual stimuli

### Surgery

Mice were initially box-anesthetized with 2% halothane in a mixture of $O_2$/$N_2O$ (50/50%). During imaging anesthesia was reduced and kept at 0.4-0.6% halothane. The animals received an injection of atropine (Franz Köhler, 0.3 mg/mouse, subcutaneously), dexamethasone (Ratiopharm, 0.2 mg/mouse, subcutaneously), and chlorprothixene (Sigma, 0.2 mg/mouse, intramuscularly) and were place in a stereotaxic apparatus. Lidocaine (2% xylocain jelly) was applied locally to all incisions. The body temperature was maintained at 37°C and heart rate was monitored throughout the experiment via attached electrocardiograph leads. An incision of the skin was made over the visual cortex and low-melting point agarose (2.5% in 0.9% NaCl) and a glass coverslip were placed over the exposed area.

### Optical imaging

Mouse visual cortical responses were recorded through the skull using the Fourier imaging technique developed by *Kalatsky and Stryker (2003)* and optimized for the assessment of ocular dominance plasticity as described by *Cang et al. (2005a)*. In this method, a temporally periodic stimulus is continuously presented to the animal and the visual cortical responses at the specific stimulus frequency are extracted by Fourier analysis. Optical images of intrinsic cortical signals were obtained by a Dalsa 1M30 CCD camera (Dalsa, Waterloo, Canada) controlled by commercially available software. Using a 135 mm×50 mm (imaging of one hemisphere) or 50 mm×50 mm (simultaneous imaging of both hemispheres) tandem lens configuration (Nikon, Inc., Melville, NY) a cortical area of 2.67x2.67 mm$^2$ or 3.6x3.6 mm$^2$ and was imaged, respectively. The surface vascular pattern and intrinsic signal images were visualized with illumination wavelengths set by a green (550 ± 10 nm) or red (610 ± 10 nm) interference filter, respectively. After acquisition of a surface image, the camera was focused 600 µm below the cortical surface. Frames were acquired at a rate of 30 Hz temporally binned to 7.5 Hz and stored as 512×512 pixel images after spatial binning of the camera image.

### Visual stimuli

Drifting horizontal bars (2° wide) were presented to the animal at a distance of 25 cm on a high refresh-rate monitor (Hitachi, ACCUVUE, HM-4921-D, 20 inches). The distance between two bars was 70° and they were presented at a temporal frequency of 0.125 Hz. For calculating ocular dominance, the visual stimulus was restricted to the binocular visual field of the left V1 (−5° to +15° azimuth, 0° azimuth corresponding to frontal direction) and animals were stimulated through either the left or the right eye in alternation. For determining the quality of retinotopic maps, we used full-field

stimulation through the contralateral eye with a horizontal (elevation maps) or vertical (azimuth map) moving bar, extending 62x92°of the visual field contralateral to the recorded hemisphere.

## Data analysis

Visual cortical maps were calculated from the acquired frames by Fourier analysis to extract the signal at the stimulation frequency using commercially available software (*Kalatsky and Stryker, 2003*). While the phase component of the signal is used for the calculation of retinotopy, the amplitude component represents the intensity of neuronal activation (expressed as fractional change in reflectance $\times 10^{-4}$) and was used to calculate OD. For that, the ipsilateral eye magnitude map was first smoothed to reduce pixel shot noise by low-pass filtering using a uniform kernel of 5x5 pixels, and then thresholded at 30% of peak response amplitude to eliminate the background noise. Then an OD-index (ODI) for every pixel in this region was calculated as: $(C-I)/(C+I)$, with C and I representing the response magnitudes of each pixel to visual stimulation of the contralateral and ipsilateral eye, respectively (*Cang et al., 2005a*). The ODI ranges from $-1$ to $+1$, with negative values representing ipsilateral and positive values representing contralateral dominance. We then computed an ODI as the average of the OD-scores of all responsive pixels (all pixels with response amplitude above 30% of peak response). Consequently, we calculated ODIs from blocks of 4 runs in which the averaged maps for each eye had at least a response magnitude of $1 \times 10^{-4}$. All ODIs of one animal (typically 3–5) were averaged for further quantification and data display. In the polar maps, hue encodes visual field position (retinotopy) and lightness encodes the magnitude of the visual responses. The quality of the retinotopic maps was assessed by the calculation described by *Cang et al. (2005b)* on contralateral eye. We selected the most responsive area (in the number of 20,000 pixels) within V1 by thresholding at 30% of peak response amplitude of the activity map. For each of the pixels within this area the difference between its visual field position and the mean position of its surrounding 24 pixels was calculated. For maps of high quality, the position differences are quite small because of smooth progression. The standard deviation of the position difference was then used as an index of the quality of retinotopic maps with small values indicating high map quality and vice versa.

## Perfusion and histology

Following optical imaging experiments, mice were deeply anesthetized with an intraperitoneal injection of 50 mg/kg pentobarbital and perfused transcardially with Phosphate-Buffered Saline (PBS) (pH 7.4, 0.1M) for 2 min followed by 4% paraformaldehyde (pH 7.4) for 8 min. The brain was removed and postfixed in 4% paraformaldehyde (pH 7.4) for one day, then transferred to a 10% sucrose solution in PBS for one day followed by a 30% sucrose solution in PBS for one or two days. The brains were frozen in methylbutane at -40°C and stored at -80°C. 40 μm thick coronal brain sections were cut on a microtome and Nissl-stained. To determine size and location of the cortical lesions every third section was analyzed under the microscope (Axioskop, Carl Zeiss) using AxioVision (AxioVs 40 4.5.0.0.). Animals in which the lesion extended into the white matter or V1 were excluded from further analyses.

## Data and statistical analyses

We used the Shapiro-Wilk test to check if the data are normally distributed within the group and the F-test to look for equality of variance within the groups. All intra- and intergroup comparisons were analyzed either by a two-tailed t-test or one-way ANOVA followed by multiple comparisons Bonferroni correction. The intergroup comparison of the enhancement of the spatial frequency, and contrast sensitivity thresholds were analyzed by two-way ANOVA with repeated measurements and Bonferroni correction. Significance levels were set as *p<0.05, **p<0.01, and ***p<0.001. Data are represented as means ± SEM.

## Acknowledgements

We thank Matthias Schink for excellent animal care and technical support, Karl-Friedrich Schmidt for help with some behavioral experiments and technical support, Alison Roland and Susanne Dehmel for comments on the manuscript and the entire Löwel-lab for inspiring discussions.

## Additional information

### Funding

| Funder | Grant reference number | Author |
|---|---|---|
| Alexander von Humboldt-Stiftung | Alexander von Humboldt-Foundation | Justyna Pielecka-Fortuna Michal G Fortuna |
| Bundesministerium für Bildung und Forschung | 01GQ0921 | Justyna Pielecka-Fortuna Evgenia Kalogeraki |
| Bundesministerium für Bildung und Forschung | 01GQ0810 | Siegrid Löwel |

The funders had no role in study design, data collection and interpretation, or the decision to submit the work for publication.

### Author contributions

JPF, Conception and design; Acquisition of data; Analysis and interpretation of data; Drafting or revising the article; EK, Acquisition of data, Analysis and interpretation of data, Drafting or revising the article; MGF, SL, Conception and design, Analysis and interpretation of data, Drafting or revising the article

### Ethics

Animal experimentation: All experimental procedures were approved by the local government (Niedersächsisches Landesamt für Verbraucherschutz und Lebensmittelsicherheit, registration number 33.9-42502-04-10/0326). All surgeries were performed under isoflurane or halothane anesthesia and every effort was made to minimize suffering.

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
