## [Decision Letter]

Thank you for submitting your work entitled "Diverging roles of matrix metalloproteinases in adult visual cortex plasticity in the healthy and lesioned brain" for peer review at *eLife*. Your submission has been favorably evaluated by a Senior editor, Matteo Carandini (Reviewing editor), and three reviewers, one of whom is Michael Stryker (Reviewer 3).

The reviewers have discussed the reviews with one another and the Reviewing editor has drafted this decision to help you prepare a revised submission.

Summary:

This study provides novel insights into the role of matrix metalloproteinases (MMPs) in visual cortex plasticity both in the normal adult brain and following a stroke. Previous evidence regarding the roles of MMPs and inflammation in the regulation of CNS plasticity has been confusing, at best, with some studies showing positive and others showing negative effects. The study reports on a series of experiments on the visual cortex of adult mice to show that blockade of MMPs prevents an important form of adult neural plasticity in otherwise intact animals, but that transient blockade of the apparently excessive MMP activity produced by an experimental stroke restores normal adult levels of plasticity in surrounding intact cortex. In other words, inhibiting MMPs prevents plasticity in healthy mice and promotes plasticity in lesioned mice.

These results reveal that MMPs have a surprisingly narrow range within which their activity permits and/or facilitates plastic changes. Thus MMP inhibitors prevent ocular dominance plasticity in normal mice and enable it following a stroke (in which MMP levels are presumed to have risen).

These results are interesting and they are compelling because they are obtained with a quantitative and sound physiological measure of visual cortical responses and a quantitative psychophysical measure of visual performance. The results are clear, and experimental and control conditions differ by many standard deviations, leaving one in no doubt about whether the results are sound. The findings are well documented and convincing.

Essential revisions:

1) The main problem with the paper lies with its conclusions. It is not clear at all that the results show "diverging roles" (as stated in the title) for the stroke and healthy situation. Rather than 'opposite' the two effects are likely to be two sides of the same coin – it appears the MMPs have to be within a narrow window, i.e. be at just at the right level: if they go down (with inhibitors) or up (after stroke) then there is no plasticity. Recasting the paper in this direction (and making it clear that MMPs are elevated by stroke, which is now implied but barely stated), would make the paper much clearer. So it would be best to change the main message of the paper and focus on the narrow window for plasticity imposed by MMPs.

2) Another problem (which becomes minor if the problem above is solved and the conclusions are recast) is that the paper compares the stroke and healthy situation with completely different time courses for the inhibitor treatment. In the healthy mice, the inhibitor is applied for the seven days of MD, in the lesioned mice; it is applied once an hour after lesioning, or twice after lesioning, in which case the plasticity is blocked. The paper says that the two effects in healthy and lesioned mice are opposite. To make this conclusion, the paper would need to examine the OD plasticity in healthy mice with a single injection of the MMP inhibitor one hour after deprivation. It is unclear if plasticity would be blocked or enhanced in this case. For a proper comparison, one should also inject the lesioned mice daily for seven days – to match the healthy mouse experiment – although it is likely that the plasticity would be gone, given that a second injection after 24 hours prevents plasticity. This would point to the fact that there are not necessarily differential effects in lesioned and healthy mice. Different time courses or doses of inhibitor action might give different results. As a consequence, as stated in point 1, it is not clear that the results show these two scenarios to be opposite. They are likely different versions of the same plasticity (dependent on the timing), which is in direct contrast to the conclusions currently made in the manuscript.

3) Also, the paper would be strengthened by considering that the 2x inhibitor treatment in stroke mice also blocks plasticity, like in the healthy MD case. The Discussion should speculate on why one inhibitor treatment enhances plasticity and more than one treatment prevents it. It seems that once again this result points to the need for MMPs to be in a narrow window.

4) Please clarify the timing of the MD relative to the lesion.

5) Is the loss of the effect of plasticity rescue after stroke due to multiple injections or the timing of the injections? If one gave only one injection at 24 hours, would the plasticity be rescued? This is important from a translational point of view. If these types of treatments would need to be given immediately, that would be important to know.

6) The behavioral tests are a strength of the paper, linking the results of MD to a behavioral output, and the results are consistent with the Prusky studies. Still, some caution should be taken when interpreting a behavioral acuity test for which we don't understand the underlying physiology. Indeed, this effect of MD has not been reproduced in the non-rodent literature. The data seem to suggest a kind of inverse amblyopia where vision gets better (as mediated through the fellow eye) after a period of monocular deprivation. This is certainly not the case in any other species (including humans) studied so far; in fact small deficits for some aspects of vision have been reported for the fellow eye. This is something to comment on at least in Discussion.

7) The Discussion is interesting, and it connects the related phenomena in a revealing and comprehensive way. It is likely to be of interest both to experimental neuroscience and to clinical research on therapy to minimize effects of stroke. However, it is lengthy, and it does not suggest or even speculate on a potential mechanism or reason for the differential effects seen in the paper. Even the elevation of MMP levels after stroke is only implied. Again, this is a problem that will disappear if the focus of the paper is recast, so that instead of focusing on divergence of the effects on healthy and stroke brain, it focuses on the need for a narrow window of MMPs.

---

## [Author Response]

Essential revisions:1) The main problem with the paper lies with its conclusions. It is not clear at all that the results show "diverging roles" (as stated in the title) for the stroke and healthy situation. Rather than 'opposite' the two effects are likely to be two sides of the same coin – it appears the MMPs have to be within a narrow window, i.e. be at just at the right level: if they go down (with inhibitors) or up (after stroke) then there is no plasticity. Recasting the paper in this direction (and making it clear that MMPs are elevated by stroke, which is now implied but barely stated), would make the paper much clearer. So it would be best to change the main message of the paper and focus on the narrow window for plasticity imposed by MMPs.

We thank the Reviewing Editor and reviewers for suggesting a better interpretation of our present results. We agree that “diverging roles” is not the best wording and interpretation of our findings. As suggested, we revised our conclusions and substantially rewrote parts of the manuscript to discuss our data in the modified interpretation framework. We are convinced that these changes significantly improved our manuscript and we believe that the recast version resolves the main problem the reviewers had: (i) our interpretation of the results; and (ii) making it clear that MMP activity is upregulated in stroke. In the following, we refer to the final text, where *major* modifications have been made:

We have changed the title from “Diverging roles of matrix metalloproteinases in adult visual cortex plasticity in the healthy and lesioned brain” to “Optimal level activity of matrix metalloproteinases is critical for adult visual plasticity in the healthy and stroke-affected brain”.

The Abstract has been modified to incorporate our new conclusion.

The Introduction has been improved to make it clear that MMPs are elevated by stroke (second paragraph). In addition, we have modified the text to accommodate the suggested new interpretation of our results.

Part of the Results section has been modified to make it clearer that MMPs are upregulated by stroke and to accommodate the additional stroke experiments (suggested in the 5^th^ comment of the reviewers).

As suggested by the Reviewing editor and reviewers in their 7^th^ comment, the Discussion section was shortened and modified according to our new interpretation of the results.

2) Another problem (which becomes minor if the problem above is solved and the conclusions are recast) is that the paper compares the stroke and healthy situation with completely different time courses for the inhibitor treatment. In the healthy mice, the inhibitor is applied for the seven days of MD, in the lesioned mice; it is applied once an hour after lesioning, or twice after lesioning, in which case the plasticity is blocked. The paper says that the two effects in healthy and lesioned mice are opposite. To make this conclusion, the paper would need to examine the OD plasticity in healthy mice with a single injection of the MMP inhibitor one hour after deprivation. It is unclear if plasticity would be blocked or enhanced in this case. For a proper comparison, one should also inject the lesioned mice daily for seven days – to match the healthy mouse experiment – although it is likely that the plasticity would be gone, given that a second injection after 24 hours prevents plasticity. This would point to the fact that there are not necessarily differential effects in lesioned and healthy mice. Different time courses or doses of inhibitor action might give different results. As a consequence, as stated in point 1, it is not clear that the results show these two scenarios to be opposite. They are likely different versions of the same plasticity (dependent on the timing), which is in direct contrast to the conclusions currently made in the manuscript.

We thank the Reviewing editor and reviewers for this important comment, which also helped us to rethink the interpretation and discussion of our results. We fully understand the reviewers concerns that our experimental design (7-day healthy mice treatment vs. 2-day treatment after stroke) did not allow us to draw the conclusion that “…the two effects in healthy and lesioned mice are opposite”. As suggested we have now modified our interpretation and substantially changed the text. We endorse the idea that in both scenarios, the role of MMPs is basically the same (enzymatic digestion of the substrate), yet the outcome of this activity is different due to altered physiological conditions. As stated in the above comment, our conclusions are now recast and we hope that this answers major reviewers concern.

3) Also, the paper would be strengthened by considering that the 2x inhibitor treatment in stroke mice also blocks plasticity, like in the healthy MD case. The Discussion should speculate on why one inhibitor treatment enhances plasticity and more than one treatment prevents it. It seems that once again this result points to the need for MMPs to be in a narrow window.

Yes, we agree with the Reviewing editor and reviewers. We therefore modified a paragraph in the Discussion, so that it addresses why one inhibitor treatment can enhance plasticity and more than one treatment prevents it (third paragraph).

4) Please clarify the timing of the MD relative to the lesion.

We apologize for not stating this important information explicitly in our manuscript. MD was performed directlyafter surgery for induction of a photothrombotic stroke, and the information is now added to the Methods section (subsection “Monocular deprivation (MD)”).

5) Is the loss of the effect of plasticity rescue after stroke due to multiple injections or the timing of the injections? If one gave only one injection at 24 hours, would the plasticity be rescued? This is important from a translational point of view. If these types of treatments would need to be given immediately, that would be important to know.

Thank you for raising this very interesting point. Inspired by your suggestion, we performed a set of additional experiments to test if treatments need to be given immediately after stroke to rescue plasticity. To this end, we induced a photothrombotic stroke in S1 (as before) of adult mice, and gave only a single injection of the MMPs-inhibitor 24h after lesion induction. Interestingly, a single injection 24h after PT also rescued plasticity (both ocular dominance- and interocular plasticity) indicating that the lost cortical plasticity after two injections is most likely due to multiple injections and *not* the timing. Importantly, these results are promising from a translational point of view as treatment did not need to start immediately after stroke induction to have a beneficial effect for rescuing lost visual plasticity. We believe that these additional findings strengthened our study and also significantly improved the interpretation of our results. We have added these important new findings to our Results section and expanded the Discussion accordingly (third paragraph).

6) The behavioral tests are a strength of the paper, linking the results of MD to a behavioral output, and the results are consistent with the Prusky studies. Still, some caution should be taken when interpreting a behavioral acuity test for which we don't understand the underlying physiology. Indeed, this effect of MD has not been reproduced in the non-rodent literature. The data seem to suggest a kind of inverse amblyopia where vision gets better (as mediated through the fellow eye) after a period of monocular deprivation. This is certainly not the case in any other species (including humans) studied so far; in fact small deficits for some aspects of vision have been reported for the fellow eye. This is something to comment on at least in Discussion.

We understand the Reviewing editor and reviewers’ concern and decided to rephrase the description of our optomotry measurements. We have now removed the expression “visual acuity” from the entire text and figures and changed it into “spatial frequency threshold of the optomotor reflex”. We hope that the modified wording now unambiguously clarifies what we have measured. Changes can be found: Introduction: third paragraph; Results: subsection “Inhibition of MMPs prevented experience-enabled improvements in visual capabilities” and subsection “Inhibition of MMPs after induction of a cortical lesion rescued experience-induced 240 improvements in visual capabilities in adult mice “; Discussion: sixth paragraph; Methods: subsection “Behavioral vision tests”; Figure legends: Figure 3; Figure 6 axis labeling for Figure 3 and Figure 6.

7) The Discussion is interesting, and it connects the related phenomena in a revealing and comprehensive way. It is likely to be of interest both to experimental neuroscience and to clinical research on therapy to minimize effects of stroke. However, it is lengthy, and it does not suggest or even speculate on a potential mechanism or reason for the differential effects seen in the paper. Even the elevation of MMP levels after stroke is only implied. Again, this is a problem that will disappear if the focus of the paper is recast, so that instead of focusing on divergence of the effects on healthy and stroke brain, it focuses on the need for a narrow window of MMPs.

Thank you for these constructive comments. We rewrote and shortened some parts of our Discussion and tried to make it clearer. We considerably changed the explanation of our findings so that the interpretation of our results is now focused on the need for a narrow window of MMP activity to promote adult visual plasticity. In addition, we made it more visible that the MMPs are upregulated after stroke (appropriate changes were made in the Introduction, Results, Discussion), and our Discussion has better interpretation of observed effects of MMP inhibition in healthy and stroke-affected mice.